# The Two-Component System 09 of *Streptococcus pneumoniae* Is Important for Metabolic Fitness and Resistance during Dissemination in the Host

**DOI:** 10.3390/microorganisms9071365

**Published:** 2021-06-23

**Authors:** Stephanie Hirschmann, Alejandro Gómez-Mejia, Thomas P. Kohler, Franziska Voß, Manfred Rohde, Max Brendel, Sven Hammerschmidt

**Affiliations:** 1Center for Functional Genomics of Microbes, Department of Molecular Genetics and Infection Biology, Interfaculty Institute for Genetics and Functional Genomics, University of Greifswald, 17487 Greifswald, Germany; hirschmans@uni-greifswald.de (S.H.); alejandro.gomezmejia@usz.ch (A.G.-M.); thomas.kohler@uni-greifswald.de (T.P.K.); franziska.voss@uni-greifswald.de (F.V.); max.brendel@uni-greifswald.de (M.B.); 2Central Facility for Microscopy, Helmholtz Centre for Infection Research, 38124 Braunschweig, Germany; manfred.rohde@helmholtz-hzi.de

**Keywords:** *Streptococcus pneumoniae*, two-component system 09, in vivo virulence

## Abstract

The two-component regulatory system 09 of *Streptococcus pneumoniae* has been shown to modulate resistance against oxidative stress as well as capsule expression. These data and the implication of TCS09 in cell wall integrity have been shown for serotype 2 strain D39. Other data have suggested strain-specific regulatory effects of TCS09. Contradictory data are known on the impact of TCS09 on virulence, but all have been explored using only the *rr09*-mutant. In this study, we have therefore deleted one or both components of the TCS09 (SP_0661 and SP_0662) in serotype 4 *S. pneumoniae* TIGR4. In vitro growth assays in chemically defined medium (CDM) using sucrose or lactose as a carbon source indicated a delayed growth of nonencapsulated *tcs09*-mutants, while encapsulated wild-type TIGR4 and *tcs09*-mutants have reduced growth in CDM with glucose. Using a set of antigen-specific antibodies, immunoblot analysis showed that only the pilus 1 backbone protein RrgB is significantly reduced in TIGR4Δ*cps*Δ*hk09*. Electron microscopy, adherence and phagocytosis assays showed no impact of TCS09 on the TIGR4 cell morphology and interaction with host cells. In contrast, in vivo infections and in particular competitive co-infection experiments demonstrated that TCS09 enhances robustness during dissemination in the host by maintaining bacterial fitness.

## 1. Introduction

As a pathobiont of the respiratory tract, *Streptococcus pneumoniae* (the pneumococcus) colonizes asymptomatically up to 60% of the human population [1]. In addition to being a natural commensal of the upper respiratory tract, the pneumococcus causes various diseases ranging from local infections with harmless clinical outcome to harmful severe invasive diseases. These include otitis media, pneumonia, meningitis and sepsis [2,3,4]. Worldwide, pneumococcal infections cause several million deaths every year, including 294,000 children under the age of five in 2015 [5]. Typically, pneumococci are transmitted via aerosols and acquired via the nasopharynx [6,7]. After overcoming the mucosal barrier, evading antimicrobial compounds, and antibodies, they have a high potential to attach to epithelial cells of the respiratory mucosa [2]. From this host compartment the bacteria gain access to the bloodstream and, depending on the route, overcome the blood–brain barrier [8,9,10].

There is a high demand on the one hand to produce virulence determinants such as the capsular polysaccharide (CPS), the toxin pneumolysin or adhesive molecules to escape the immune system and to facilitate colonization or interaction with host molecules or cellular structures [3,11,12]. On the other hand, pneumococcal fitness has been shown to be crucial for colonization and invasive disease [13,14]. The adaptation of pneumococcal metabolism to the conditions of the conquered host compartments is pivotal to maintain fitness and survival for a successful colonization and invasion [13,14,15]. However, the expression of virulence factors and genes that contribute to bacterial fitness correlate with the stage of infection and is, in many cases, triggered by the response to environmental factors [3,12,16]. When this signaling is mediated by two-component regulatory systems (TCS) the external stimulus is sensed by a membrane-bound receptor, typically a histidine kinase, which autophosphorylates and transmits the signal to the intracellular regulatory protein [17,18,19].

Similar to other human pathogenic bacteria, pneumococcal TCSs are directly associated with adaptation to host compartments or the virulence properties of *S. pneumoniae* [13,17,20,21,22]. Pneumococci express 13 TCSs and one orphan response regulator. Their role in pneumococcal physiology, fitness and virulence has been extensively discussed recently [13,17,23]. For TCS09, the impact on virulence has already been demonstrated by several studies [11,24,25,26]. The deletion of the RR09 attenuated virulence of serotype 2 strain *S. pneumoniae* D39 in a mouse pneumonia and bacteremia model, whereas virulence of the *rr09*-mutant of pneumococcal serotype 3 strain 0100993 was unaffected [24]. For TIGR4, the strain that is used in this study, RR09 has been shown to contribute to the development of pneumonia but not bacteremia [11]. Thus, TCS09 has a strain- and serotype-specific impact on pneumococcal metabolic processes and virulence. However, the environmental stimulus, the target genes and the reasons for the strain-specific effects have not yet been explored.

In a previous study, we deciphered the role of TCS09 for *S. pneumoniae* D39 metabolic fitness, autolysis and oxidative stress. Using transcriptomics, we indicated differential gene expression of genes involved in carbohydrate metabolism and illustrated an altered CPS amount and phenotype in *tcs09*-mutants [23]. Because of these changes, we hypothesized that the TCS09 contributes at least indirectly to pneumococcal colonization and virulence. So far, the individual role of all TCS09 components (RR09, HK09 and both) on the phenotype and virulence in TIGR4 has not been investigated. In this study, we report the impact of RR09 and HK09 and the complete TCS09 on the pathophysiology and virulence of *S. pneumoniae* TIGR4 by growth experiments, in vitro adherence and phagocytosis assays, and finally, by in vivo mouse infection assays. Our data suggest that TCS09 is crucial for full fitness and colonization under in vivo conditions.

## 2. Materials and Methods

### 2.1. Bacterial Strains and Culture Conditions

Encapsulated and nonencapsulated (Δ*cps*) *Streptococcus pneumoniae* serotype 4 (TIGR4) parental strains and isogenic *tcs09*-mutants (Table 1) were used in this study. Pneumococci grown on blood agar plates (Oxoid) with appropriate antibiotics were inoculated in chemically-defined medium (CDM) RPMI_modi_ (RPMI1640: GE Healthcare; RPMI_modi_: [14]) supplemented with 1% *w*/*v* carbon source (glucose, sucrose or lactose) or Todd-Hewitt-Broth (Roth) supplemented with 0.5% yeast extract. The bacteria were cultivated at 37 °C without agitation up to the early logarithmic (OD_600nm_ 0.35) or middle logarithmic phase (OD_600nm_ 0.6). For growth analysis, the data were plotted linearly against time. The doubling time of the strains was calculated as follows:(1)$$\mu \;[min^{-1}]\;=\;\frac{\left((\ln X\;\left(t_{2}\right)-\ln X\;\left(t_{1}\right)\right)}{\left(t_{2}-t_{1}\right)}$$
(2)$$g\;\left[min\right]\;=\frac{\ln2}{\mu }$$
μ = growth rate*X* (*t*_1_) = cell density at time *t*_1_*X* (*t*_2_) = cell density at time *t*_2_*g* = doubling time

### 2.2. Generation of Pneumococcal Mutants

Pneumococcal single mutants (RR09 or HK09) or double mutants (RR09 and HK09), were generated by insertion-deletion mutagenesis in *S. pneumoniae* TIGR4, TIGR4Δ*cps* and TIGR4*lux* using the protocol and plasmid constructs described in a previous study [23].

### 2.3. Immunoblot Analysis of Whole Bacterial Lysates Using the LI-COR Technology

Nonencapsulated *S. pneumoniae* TIGR4Δ*cps* and isogenic *tcs09*-mutants were cultured to an OD_600nm_ of 0.6 in glucose supplemented CDM in triplicates. Subsequently, the cultures were centrifuged for 6 min at 3200× *g* at room temperature (RT) and the bacterial pellet was resuspended in PBS (pH 7.4). A concentration of 2 × 10^8^ bacteria was loaded onto a 12% SDS-PAGE and then blotted onto a nitrocellulose membrane by semidry blotting. The membrane was blocked for 2 h at RT using 5% skim milk (Roth) in TBS, pH 7.4. Incubation with mouse polyclonal antibodies (1:500 in 5% skim milk + TBS/0.01% Tween) against selected proteins was performed overnight at 4 °C. Membranes were then washed with TBS/0.01% Tween, and detection of various pneumococcal proteins was performed using a secondary fluorescent labeled IRDye^®^ 800CW (goat-anti-mouse IgG; 1:15,000 in 5% skim milk in TBS/0.01% Tween). A rabbit polyclonal anti-enolase antibody (1:12,500 in 5% skim milk + TBS/0.01% Tween) was used as a loading control and detected using fluorescent labeled IRDye^®^ 680RD goat-anti-rabbit IgG. The primary antibodies were detected by incubation with the appropriate antibodies for 60 min in the dark at RT. Membranes were washed with TBS/0.01% Tween and finally twice with TBS. Scanning of the membranes was performed using an Odyssey^®^CLx (LI-COR) scanner.

### 2.4. RNA Purification and Quantitative Real-Time PCR (qPCR)

RNA isolation, purification and qPCR were performed as described [23]. In brief, pneumococcal strain TIGR4Δ*cps* and isogenic ∆*rr09*-, ∆*hk09*- and ∆*tcs09*-mutants were cultivated in glucose supplemented CDM up to an OD_600nm_ of 0.6. Ice cold killing buffer was added to the bacterial cultures and centrifuged, and then the supernatant was removed. Bacterial pellets were immediately frozen in liquid nitrogen and stored at −80 °C. For total RNA isolation, samples were treated with acidic phenol-chloroform, subjected to TURBO™ DNase (2 U/reaction; Invitrogen) digestion and purified using RNA cleanup and concentration kit (Norgen Biotek Corp., Thorold, ON, Canada). The amount of RNA was measured using a NanoDrop ND-1000 spectrophotometer (Peqlab, Erlangen, Germany). Isolated RNA was transcribed in cDNA using Superscript III reverse transcriptase (Thermofisher, Bonn, Germany) and random hexamer primers (BioRad, Feldkirchen, Germany). A StepOnePlus thermocycler (Applied Biosystems, Darmstadt, Germany) and the iTaq Universal SYBR Green Supermix (BioRad, Feldkirchen, Germany) were used for quantitative real-time PCR. Enolase (*sp_1128* in TIGR4) was used as reference gene. The *rrgA* (*sp_0462*), *rrgB* (*sp_0463*) and *rrgC* (*sp_0464*) genes were used as targets (Primer list Table 2). The qPCR conditions using 20 ng/μL cDNA as template were as follows: initial denaturation at 95 °C for 2 min, denaturation at 95 °C for 15 sec, primer annealing at 60 °C for 30 sec and elongation at 72 °C for 30 sec for 40 cycles with a final melting curve step for quality control. Differential gene expression was calculated by the ΔΔCt method.

### 2.5. Field Emission Scanning Electron Microscopy (FESEM) and Transmission Electron Microscopy (TEM)

FESEM and TEM were performed to study the cell morphology and capsule of pneumococcal wild-types TIGR4, TIGR4Δ*cps* and their respective isogenic *tcs09*-mutants. Bacteria were cultivated in CDM with glucose until OD_600nm_ 0.3 (encapsulated strains) or 0.6 (nonencapsulated strains) at 37 °C. Samples were prepared for FESEM and TEM as described [23]. In brief, encapsulated strains were fixed first with 2.5% glutardialdehyde, 2% paraformaldehyde, 0.075% ruthenium red and 75 mM L-lysine acetate salt and stained with 1% osmium solution containing ruthenium red. In contrast, nonencapsulated pneumococci were fixed with 2% glutardialdehyde and 5% paraformaldehyde. For FESEM, samples were placed on coverslips, fixed with 1% glutaraldehyde and dehydrated in a graded series of acetone (10–100%). Samples were then subjected to critical-point-drying with liquid CO_2_ and covered with a gold-palladium film by sputter coating. Processed samples were examined in a field emission scanning electron microscope (Zeiss Merlin) using the HESE2 Everhart Thornley SE detector and the in-lens SE detector in a 75:25 ratio with an acceleration voltage of 5 kV. For TEM, bacteria were fixed as described above, mixed with an equal volume of 2% water agar solidified, cut and dehydrated in a graded series of ethanol (10–50%) and incubated in 70% ethanol with 2% uranylacetate. Samples were then dehydrated and infiltrated with aromatic acrylic resin LRWhite. Ultrathin sections were cut, counterstained with 4% aqueous uranyl acetate and examined in a Zeiss TEM 910 transmission electron microscope at an acceleration voltage of 80 kV.

### 2.6. Infection of Epithelial Cells for Pneumococcal Adherence Analysis by Immunofluorescence Microscopy

Epithelial A549 cells (ATCC CCL-185) were seeded at 5 × 10^4^ cells per well in 24-well tissue culture plates (Greiner Bio One, Frickenhausen, Germany) on 12 mm diameter glass coverslips and cultured for 2 days at 37 °C and 5% CO_2_ in DMEM High Glucose (HyClone™, Freiburg, Germany) supplemented with 10% (*v*/*v*) heat-inactivated fetal bovine serum (FBS) (Gibco, Bonn, Germany), 4 mM glutamine and 1 mM sodium pyruvate. The cells were washed three times with DMEM containing 1% heat-inactivated FBS (infection medium). Cells were infected with TIGR4Δ*cps* and isogenic *tcs09*-mutants using a multiplicity of infection (MOI) of 25 pneumococci per cell. Pneumococci were grown in THY medium on the day of infection to OD_600nm_ 0.35 and adjusted to an MOI 25 in infection medium. Infections were performed at 37 °C and 5% CO_2_ for 3 h. After 3 h, non-adherent bacteria were removed by washing three times with PBS. Cells were fixed with 1% paraformaldehyde overnight at 4 °C and subsequently used for immunofluorescence staining. For double-immunofluorescence (DIF) staining, the coverslips were carefully removed from the wells and washed three times with PBS. To avoid nonspecific binding of the pneumococcal antibody, cells were blocked with 10% FBS in PBS (3 h, RT). Visualization of extracellular adherent pneumococci was performed with pneumococcal-specific polyclonal antibodies (1:500, in PBS/10%FBS) for 30 min at RT. After washing the coverslips three times with PBS, bacteria were stained with Alexa 568-coupled goat anti-rabbit IgG (abcam, Cambridge, UK) (30 min, 1:500, in PBS/10% FBS). Coverslips were then washed again three times with PBS. Visualization of the actin cytoskeleton was conducted after permeabilization of the cells with 0.1% Triton X-100 (10 min) and incubation with Phalloidin-iFluor™-488 conjugate (abcam, UK) (1:2000, in PBS/10% FCS) for 30 min at RT. After the coverslips were repeatedly washed three times in PBS, they were fixed on a slide using Mowiol. Image acquisition was performed with a fluorescence microscope (Zeiss Axiovert, Jena, Germany) and VisiView^®^ software (Visitron Systems GmbH, Puchheim, Germany).

### 2.7. Quantification of Phagocytosed Pneumococci by the Antibiotic Protection Assay

In a 24-well cell culture plate, 5 × 10^4^ J774 cells (ATCC TIB-67) per well were seeded for triplicates, cultured in RPMI1640 with L-glutamine supplemented with 1 mM sodium pyruvate and 10% (*v*/*v*) heat-inactivated FBS for 48 h at 37 °C with 5% CO_2_. On the day of infection, J774 cells were washed three times with infection medium (RPMI1640 + 1% FBS). Cultivation of TIGR4Δ*cps* and isogenic *tcs09*-mutants was performed in THY to an OD_600nm_ of 0.35. The infection dose was set to a MOI of 50. Subsequently, the cell culture plate was centrifuged at 9× *g* for 2 min to allow contact of the bacteria with the phagocytes and incubated at 37 °C and 5% CO_2_ for 30 min. After infection, the cells were washed three times with infection medium. To kill extracellular pneumococci, J774 cells were incubated in infection medium supplemented with gentamicin (100 μg/mL) and penicillin G (300 U/mL) for 1 h at 37 °C and 5% CO_2_. Cells were washed again three times with infection medium, and subsequently or 1–3 h later, J774 cells were lysed by adding 1% saponin (in infection medium) at 37 °C and 5% CO_2_ for 10 min, resulting in the release of intracellular pneumococci. The number of recovered bacteria was determined by plating the lysate on blood agar plates, and after overnight incubation at 37 °C and 5% CO_2_, colony forming units (CFU) were enumerated by counting. Experiments were performed at least three times in triplicates.

### 2.8. Quantification of Phagocyte-Associated and Intracellular Pneumococci by Immunofluorescence Microscopy

J774 cells were seeded (5 × 10^4^ cells/well) in a 24-well cell culture plate on 12 mm sterile glass coverslips. Cultivation and preparation of the cells for infections was performed as described above. After 30 min infection with pneumococci, each well was carefully washed with infection medium. J774 cells and pneumococci were then fixed with 1% paraformaldehyde in PBS overnight at 4 °C. The coverslips were removed and washed three times with PBS followed by blocking with 10% FBS in PBS (3 h, RT). Subsequently, the cells were washed three times with PBS, followed by an incubation with the pneumococcal-specific antibody (1:500 in PBS/10%FBS) for 30 min at RT. Another wash step (3× in PBS) was followed by staining with the secondary Alexa 488-coupled goat anti-rabbit IgG (abcam) (30 min, 1:500 in PBS/10% FBS). The visualization of the internalized bacteria was done after Triton X-100 conveyed permeabilization of the cells (0.1%) and after renewed incubation with the rabbit anti-pneumococci IgG (30 min, 1:500) and the Alexa 568-coupled goat anti-rabbit IgG (30 min, 1:500). After incubation was complete, the coverslips were washed again three times in PBS and finally fixed on a slide using Mowiol. Image acquisition was performed with a fluorescence microscope (Zeiss Axiovert) and VisiView^®^ software (Visitron Systems GmbH).

### 2.9. Acute Pneumonia Infection Model

For acute pneumonia, 8–10-week-old female CD-1 mice (Charles River) (*n* = 10) were intranasally infected with the bioluminescent wild-type strain TIGR4*lux* or isogenic *tcs09*-mutants [14]. Bacteria were cultured in THY supplemented with 10% heat-inactivated FBS to an OD_600nm_ 0.35. After centrifugation at 3200× *g* for 10 min, an infection dose of 1 × 10^8^ CFU in 10 μL was prepared. Prior to intranasal infections, mice were anesthetized by an intraperitoneal injection of a mixture of ketamine (Ketanest S) and xylazine (Rompun^®^) according to their average weight; 90 U/10 μL of hyaluronidase was added per infection dose. The final infection dose (20 μL) was added dropwise to the nostrils of the mice for intranasal infection. The infected mouse was further held up for a brief moment, preventing the inoculum from receding. Animals were observed daily for weight. Real-time visualization and documentation of the spread of bioluminescent pneumococci was performed using IVIS^®^ Spectrum Imaging System (Caliper Life Sciences). The first measurement was taken approximately 24 h after infection followed by 8 h intervals. During the measurements, the animals were anesthetized with isoflurane and bioluminescence was measured for 1 min with medium “binning” factor. The emitted photons were recorded and quantified using LivingImage^®^ 4.1 software (Caliper Life Sciences).

### 2.10. Systemic Infection Model

To investigate the systemic course of infection, 8–10-week-old female CD-1 mice (Charles River) (*n* = 8) were infected intraperitoneally with 1 × 10^4^ CFU wild-type bioluminescent strain TIGR4*lux* or isogenic *tcs09*-mutants in 200 μL PBS. Post infection, the weight and severity of disease of infected mice were monitored at 8 h intervals starting 24 h post infection. Animals were observed daily for clinical score monitoring.

### 2.11. Co-Infection Model

Pneumococci were cultured in THY supplemented with 10% heat-inactivated FBS to an OD_600nm_ 0.35. In a competition infection experiment, 8–10-week-old female CD-1 mice (Charles River) (*n* = 10) were intranasally infected with 1 × 10^7^ CFU of bioluminescent wild-type TIGR4*lux* and 1 × 10^7^ CFU of one of the *tcs09*-mutant strains in a 1:1 ratio. The CFU was determined in the nasopharynx, bronchi, blood, lung tissue and brain after intranasal infection 24 h and 48 h post infection. Briefly, mice were sacrificed, blood was taken from the heart and the trachea was dissected for nasal and bronchoalveolar lavage, and 1 mL of sterile PBS was passed through the nasopharynx or inserted into the lungs with the inserted trachea cannula and collected after passage. Afterwards, lung tissue and brain were removed and homogenized in 1 mL PBS. The output of mutant versus wild-type bacteria was determined on selective blood agar plates containing kanamycin and/or erythromycin. The competitive index (CI) was calculated. A value of 1 indicates identical output CFU of wild-type and mutant bacteria, while CI values < 1 indicate a higher output of wild-type bacteria and values > 1 a higher output of mutant bacteria.

### 2.12. Statistical Analysis

Unless stated otherwise, all the data collected in this study are presented as mean of at least three independent experiments with the standard deviation ± SD. The results were statistically evaluated using a two-way Anova or the unpaired two-tailed student’s *t*-test (GraphPad Prism 5.01). A *p*-value < 0.05 was considered as statistically significant. All Kaplan–Meier survival curves were compared using the Log-rank-test.

## 3. Results

### 3.1. Growth Analysis of Encapsulated and Nonencapsulated Wild-Type TIGR4 and tcs09-Mutants under Nutrient-Defined Conditions

To assess how the TCS09 affects *S. pneumoniae* fitness when various carbon sources are provided, we cultured encapsulated and nonencapsulated wild-type strains TIGR4, TIGR4Δ*cps* and isogenic *tcs09*-mutants in chemically defined medium RPMI_modi_ in the presence of different carbon sources. In glucose-supplemented CDM, the encapsulated TIGR4 strains showed only moderate growth and reached the stationary phase already after 4 h. Growth of the mutants TIGR4Δ*rr09* (*g* = 121 min), TIGR4Δ*hk09* (*g* = 457 min) and TIGR4Δ*tcs09* (*g* = 133 min) was significantly slower than growth of the wild-type TIGR4 (*g* = 106 min) (Figure 1A–C). In contrast, growth kinetics of nonencapsulated wild-type TIGR4Δ*cps* and isogenic *tcs09*-mutant strains showed growth curves with a short lag phase of 1 h and an exponential phase of 6 h, followed by a stationary phase after approximately 7 h (Figure 1D–F). The three nonencapsulated *tcs09*-mutants had a slight delay in growth compared to the wild-type, with an average doubling time of 99 min versus 81 min, respectively. For TIGR4, TIGR4Δ*cps* and their isogenic *tcs09*-mutants, we did not observe bacterial cell lysis immediately after reaching the stationary phase. When we used disaccharides such as sucrose and lactose as carbon sources, the three nonencapsulated *tcs09*-mutants showed an extended lag phase compared to the parental TIGR4Δ*cps* (Figure 1G–L). Growth of TIGR4Δ*cps*Δ*rr09* (*g* = 70–98 min), TIGR4Δ*cps*Δ*hk09* (*g* = 76–101 min) and TIGR4Δ*cps*Δ*tcs09* (*g* = 70–96 min) was slightly slower than growth of wild-type TIGR4Δ*cps* (*g* = 57–75 min). A stationary phase was not observed; instead, the nonencapsulated parental strain TIGR4Δ*cps* and isogenic *tcs09*-mutants started to lyse immediately after reaching their maximum optical density, which was measured after 6 h and 8 h (sucrose) or 9 h (lactose). Importantly, the maximum cell density was twice as high as the cultures in which glucose was used as a carbon source (Figure 1G–L).

### 3.2. Expression of Important Virulence Factors in TIGR4Δcps Wild-Type and Isogenic tcs09-Mutants

A strain-dependent effect on gene regulation of phosphotransferase systems (PTS) and enzymes of carbohydrate metabolism was indicated for *rr09*-mutants, when D39 and TIGR4 were compared [25]. RR09 is associated with pili regulation, whereas no other important virulence factors have yet been found to be differentially regulated by RR09 [25]. In a *hk09*-mutant of D39, *pspA* and *phtD* encoding important adhesins as well as the *aga* operon involved in galactose utilization were shown to be upregulated [23], highlighting the importance of screening *rr09*-, *hk09*- and *tcs09*-mutants for differential gene and protein expression. Hence, we have investigated the expression of important proteins for pneumococcal fitness and virulence by immunoblot analysis. We have chosen representative candidates of lipoproteins, choline-binding proteins, sortase-anchored proteins and intracellular proteins for a quantitative analysis. For this purpose, we cultured nonencapsulated *tcs09*-mutants and the parental strain TIGR4Δ*cps* in glucose-supplemented CDM to an OD_600nm_ of 0.6 and applied 2 × 10^8^ cells for protein expression. Our immunoblot analysis suggested a lower expression of RrgB in the *hk09*-mutant, which is the backbone protein of type 1 pilus (Figure 2A,B, Appendix A). Apart from this, none of the other proteins showed a differential protein expression. To validate the differential expression of RrgB and Pilus-1 expression in TIGR4Δ*cps*Δ*hk09*, we applied qPCR. We therefore isolated RNA from TIGR4Δ*cps* wild-type bacteria and isogenic *tcs09*-mutants grown in CDM with glucose as carbon source. The qPCR results revealed a significant upregulation of *rrgA*, *rrgB* and *rrgC* in *rr09*- and *hk09*-mutants, which is not in accordance with our protein expression analysis (Figure 2C). Thus, while the TCS09 seems to have a regulatory effect on Pilus-type 1 expression, the regulatory effect remains still unclear due to the difference in data at the protein level and transcriptome level.

### 3.3. Analysis of Pneumococcal Morphology and Capsule Content

As mentioned, we have observed changes in the cell morphology of pneumococcal D39 mutant lacking components of the TCS09 [23]. We therefore investigated as part of phenotypic characterization the influence of TCS09-deficiency on TIGR4 cell morphology, cell division and capsule amount. Encapsulated and nonencapsulated TIGR4 and TIGR4Δ*cps*, as well as the isogenic *tcs09*-mutants, were cultured in CDM with glucose as carbon source. We applied field emission scanning electron microscopy (FESEM) and transmission electron microscopy (TEM) to illustrate potential alterations. FESEM and TEM images of the isogenic *tcs09*-mutants and the wild-type strains TIGR4 and TIGR4Δ*cps* showed no morphological differences in cell shape or changes in cell division (Figure 3). In addition to cell shape, size and arrangement of septa, we further examined *tcs09*-mutants for alterations in capsule structures or capsule amount. To visualize the effect of TCS09-deficiency on the capsule amount, we used the lysine-ruthenium red (LRR) treatment to preserve the CPS. In contrast to *tcs09*-mutants of *S. pneumoniae* D39 from our previous study, all TIGR4 *tcs09*-mutants showed capsule structures that were similar to the parental TIGR4 strain (Figure 3B). Although the cytoplasm of all *tcs09*-mutants of nonencapsulated strains exhibit more distinct small white areas, suggesting alterations in cell morphology, we could not detect vesicle formation in TIGR4-mutants (Figure 3) that were recently indicated for *tcs09*-mutants of strain D39 [23]. In conclusion, the TCS09 of TIGR4 has little, if any, effect on pneumococcal cell morphology and capsule layer, suggesting the effects of TCS09 on cellular processes being strain-specific.

### 3.4. Role of the TCS09 on Pneumococcal Adherence

To investigate the effect of loss of function of TCS09 on pneumococcal adherence, we infected human lung epithelial cells (A549) with the nonencapsulated *S. pneumoniae* strain TIGR4Δ*cps* and isogenic *tcs09*-mutants for 3 h. We quantified the number of adherent pneumococci of the different strains by counting host cell attached bacteria after immunofluorescence staining. The *tcs09*-mutants did not show significant changes in their capacity to interact with A549 epithelial cells 3 h post infection compared to the parental strain TIGR4Δ*cps* (Figure 4). The adherence data suggest that under this selected condition TCS09 does not significantly affect the interaction of pneumococci with epithelial cells.

### 3.5. Impact of the Pneumococcal TCS09 on Uptake by Phagocytes

A critical step in pneumococcal invasive disease is the protection against phagocytosis by innate immune cells. To decipher whether the TCS09 of *S. pneumoniae* TIGR4 affects uptake by phagocytes and intracellular killing, we investigated uptake of TIGR4Δ*cps* and isogenic *tcs09*-mutants in in vitro infection experiments using murine J774 macrophages. Double-immunofluorescence staining was performed to visualize extracellular and intracellular pneumococci and to quantify associated and intracellular bacteria (Figure 5A). Thirty minutes post-infection uptake of the *tcs09*-mutants by phagocytes was similar compared to wild-type TIGR4Δ*cps* (Figure 5B). In accordance with these data, the antibiotic protection assay, which is used to quantify intracellular survivors, confirmed the results of the immunofluorescence microscopy (Figure 5C). Both approaches showed that uptake and intracellular survival of pneumococci is not altered in the absence of a functional TCS09. We therefore tested whether the intracellular survival of *tcs09*-mutants is affected over time compared to the parental strain TIGR4Δ*cps*. In kinetic experiments, intracellular pneumococci were isolated over 3 h. Intracellular bacterial survivors were recovered from macrophages after incubation for 1, 2 or 3 h post-killing of extracellular pneumococci by antibiotic treatment. In general, pneumococci are killed intracellularly in macrophages. This has been described earlier and is independent of the pneumococcal strain [28,29]. Interestingly, 1 h post incubation, killing of TIGR4Δ*cps*Δ*rr09* and TIGR4Δ*cps*Δ*hk09* was significantly delayed compared to the wild-type strain (Figure 5D). At this time point, the survival rate of these two mutants and TIGR4Δ*cps*Δ*tcs09* was significantly higher compared to TIGR4Δ*cps*. However, 1 h later and 2 h post incubation, the killing rate of the RR09- and HK09-deficient mutants reached the levels of the parental strain. Finally, 3 h post incubation, only the double mutant TIGR4Δ*cps*Δ*tcs09* was significantly better killed than the other mutants or parental TIGR4Δ*cps* strain. Although the intracellular fate of *tcs09*-mutants is moderately changed, the overall effect of the TCS09 on TIGR4 uptake by professional phagocytes and intracellular survival seems to be of minor importance.

### 3.6. Influence of TCS09 on Lung Infections Caused by S. pneumoniae TIGR4

We further employed the acute pneumonia mouse infection model to investigate the role of TCS09 on the pathophysiology of TIGR4 under in vivo conditions. We visualized the progression of pneumococcal disease by in vivo bioimaging of mice intranasally infected with bioluminescent TIGR4*lux* or its isogenic *tcs09*-mutants. In addition, we monitored the severity of disease and mouse survival rates, which are illustrated in a Kaplan–Meier diagram (Figure 6). Mice infected individually with wild-type TIGR4*lux*, or the mutants TIGR4*lux*Δ*rr09*, TIGR4*lux*Δ*hk09* or TIGR4*lux*Δ*tcs09,* survived for at least 32 h (Figure 6B). Dissemination of pneumococci from the nasopharynx into the lungs and blood occurred independently from the genetic background of the strain as indicated by the bioluminescence (Figure 6A). First signs of pneumococci in the lungs could be detected already 24 h post infection, and disease progression was similar for all strains. After 88 h, 80% of mice infected with TIGR4*lux*Δ*hk09* died, whereas this was the case after 112 h or 144 h for mice infected with the wild-type TIGR4*lux* and TIGR4*lux*Δ*tcs09*, or TIGR4*lux*Δ*rr09*, respectively. These results suggest that the virulence potential of TIGR4 is not modulated by the TCS09 in vivo.

### 3.7. Impact of TCS09 on Virulence in the Systemic Mouse Infection Model

To analyze the effect of TCS09 deficiency on pneumococcal induced sepsis, we infected CD-1 mice intraperitoneally with pneumococcal strains TIGR4*lux*, TIGR4*lux*Δ*rr09*, TIGR4*lux*Δ*hk09* or TIGR4*lux*Δ*tcs09*. The survival rate of infected mice was monitored. The results are represented in a Kaplan–Meier diagram (Figure 7A). All mice, independently of the mutant, showed no clinical sign of disease and survived for at least 40 h. After 48–64 h, monitoring of infected mice indicated severe signs of disease. The exceptions were each mouse infected with the *hk09*- or *tcs09*–mutant surviving for 120 h (Figure 7). Similar to the acute pneumonia infection model, the systemic mouse infection model revealed no statistical differences in the survival rates between mice infected with the wild-type strain and one of the *tcs09*-mutants. Thus, TCS09 deficiency has no effect on pneumococcal-induced sepsis.

### 3.8. Importance of TCS09 for the Bacterial Load in the Respiratory Tract after Co-Infection

The results so far suggested that TCS09 of TIGR4 does not dramatically affect virulence during pneumonia and invasive pneumococcal disease. However, the impact on pneumococcal fitness in the respiratory tract and dissemination has not been addressed in a competitive manner. We have therefore analyzed the role of TCS09 on pneumococcal fitness using the competitive colonization and infection model. Mice were intranasally co-infected with equal CFUs of *S. pneumoniae* TIGR4*lux* and TIGR4*lux*Δ*rr09* or TIGR4*lux*Δ*hk09* or TIGR4*lux*Δ*tcs09*. We then determined the bacterial load 24 h and 48 h post infection by recovering the bacteria from the nasopharynx, respiratory airways, blood, lung and brain. Finally, competitive indexes (CIs) were calculated using the determined CFU of the *tcs09*-mutants and wild-type TIGR4*lux*. At 24 h and 48 h post co-infection, the bacterial loads in the nasopharynx (NP) showed no significant differences between wild-type TIGR4*lux* and *tcs09*-mutants (Figure 8, Appendix A). The *rr09*-mutant was out-competed by the wild-type TIGR4*lux* in the blood 24 h post infection (CI: 0.47) and remained low 48 h (CI: 0.57) post infection (Figure 8A, Appendix A). After 48 h, TIGR4*lux*Δ*rr09* was also present with a significantly lower number of bacteria in the brain compared to the wild-type TIGR4*lux* (CI: 0.48) (Figure 8A, Appendix A). The co-infection and competitive infections further showed that the TIGR4*lux*Δ*hk09* was out-competed by the wild-type bacteria in the pulmonary bronchi (CI: 0.38) and lung tissue (CI: 0.35) 48 h post infection (Figure 8B, Appendix A). Interestingly, the observed effect was more pronounced with TIGR4*lux*/TIGR4*lux*Δ*tcs09*. Here, the double mutant TIGR4*lux*Δ*tcs09* was already out-competed by the wild-type TIGR4*lux* in the lung tissue (CI: 0.48) after 24 h (Figure 8C, Appendix A). After 48 h, this mutant was additionally out-competed by the wild-type bacteria in the blood (CI: 0.45) and brain (CI: 0.41) (Figure 8C, Appendix A). These results suggest that the regulatory function of TCS09 under in vivo conditions is required to maintain physiological fitness and robustness, which in turn allows a higher efficiency to initiate invasive infections (Figure 8).

## 4. Discussion

In this study, we have explored the role of *S. pneumoniae* TIGR4 TCS09, which is also referred to as ZmpRS [11] on bacterial fitness and virulence by applying relevant culture experiments and mouse infection models. The in vivo infection models included the acute pneumonia, sepsis and a co-infection model. To monitor disease progression, we imaged the dissemination and localization of bacteria in mice using real-time bioluminescent parental TIGR4*lux* and *tcs09*-mutants.

Initial results of growth experiments indicated a lower fitness of *tcs09*-mutants depending on the state of encapsulation and carbon source provided in CDM. Using glucose, the main carbon source in the blood [30], only the encapsulated wild-type TIGR4 and isogenic *tcs09*-mutants showed a substantial growth defect in CDM, which was even more pronounced in the mutant strains (Figure 1A–C). The reason for the reduced fitness has to be explored further, but we hypothesize that required nutrients for high-energy-demanding capsule production are limited for the mutants when cultured in CDM. As shown by in vitro proteome analysis, limited iron availability in CDM leads to a decreased amount of uridine diphosphate-*N*-acetyl-D-mannosamine, which is an important precursor for capsule biosynthesis [31]. In our previous TCS09 study with serotype 2 strain D39, we monitored an indirect influence of TCS09 on capsule expression, which reduced the growth rate in glucose-supplemented CDM [23]. Hence, the similar observation for TIGR4 with increased doubling time strengthens the hypothesis that TCS09 contributes to capsule modulation in pneumococci.

However, the level of glucose is limited in the nasopharynx, therefore pneumococci use di- and monosaccharides cleaved from glycoproteins, including lactose, fructose, sucrose, mannose, sialic acid and galactose, as their main energy source [32,33,34]. Uptake of lactose occurs via the lactose-specific PTS IIABC (SP_0476, SP_0478, SP_1185 and SP_1186) transporter systems, and the resulting lactose-6-phosphate is converted in several metabolic steps in glyceraldehyde-3-phosphate and dihydroxyacetone, both intermediates of the glycolysis [32,35,36,37,38]. Sucrose is a disaccharide of glucose and fructose that is taken up by the sucrose-specific PTS IIBCA components (SP_1722), phosphorylated to sucrose-6-phosphate, and enters glycolysis as glucose-6-phosphate [39]. Growth analyses with lactose and sucrose revealed that *S. pneumoniae* can utilize sucrose and lactose equally, whereas the *tcs09*-mutants showed an extended lag phase (Figure 1G–L). In a previous study, we demonstrated a link between TCS09 and carbohydrate metabolism in D39. We indicated downregulation of the regulator AgaR in mutants deficient for the TCS09 system, which in turn leads to an upregulation of the *aga* operon involved in galactose and galactosamine metabolism [23]. For both encapsulated as well as nonencapsulated strains, no growth defects were observed in complex medium THY, suggesting that e.g., oligopeptides or other components compensate for the regulatory deficits induced by the *tcs09*-mutation (Appendix A).

Similar to D39 [23], we also investigated the TIGR4 cell morphology of *tcs09*-mutants. However, in contrast to D39, our FESEM and TEM did not show dramatic alterations of the cell morphology and altered septum formation of generated *tcs09*-mutants in TIGR4 and TIGR4Δ*cps* as we have visualized for D39 mutants (Figure 3). LRR staining of pneumococci enabling illustration of CPS also showed no visible alterations in the CPS number of *tcs09*-mutants. Thus, the regulatory impact of TCS09 on pneumococcal cell morphology and capsule amount is a strain-specific effect, and current data suggest that TCS09 has only a minor influence on the TIGR4 morphology under in vitro conditions when compared to D39 pneumococci [23].

Successful colonization of the host is an important step to disseminate in submucosa tissues during invasive infections. For adhesion to and invasion into host cells, pneumococci produce enzymes unmasking receptors and adhesins such as PspC, Pilus-1 and PavB that facilitate the interaction with host cell receptors [16,40,41]. In the *hk09*-mutant, the expression of the pilus backbone RrgB protein is reduced at the protein level. However, on the mRNA level, we measured an upregulation of *rrgB* expression in the *hk09*- and even more in the *rr09*-mutant of TIGR4. Interestingly, these contradictory data fit perfectly with an earlier study showing that the regulation of the *rlrA* pathogenicity island, which consists of all pilus-type 1 genes including *rrgB*, is strongly dependent on the medium in which the *rr09*-mutant was cultured. In addition, up- or down-regulation measured in the transcriptome profiles was dependent on the optical densities of pneumococci [25,42]. It is noteworthy that our whole bacterial cell lysates and mRNA were prepared from bacteria cultured in CDM under identical conditions. Our data and the data of the other study suggest an influence of TCS09 on adherence or colonization. However, our in vitro cell-culture-based infection experiments revealed no significant difference between the parental TIGR4Δ*cps* and *tcs09*-mutants (Figure 4). Thus, the amount of type 1 pilus on *tcs09*-mutants seems on the one hand to be sufficient for pneumococcal adherence and on the other hand not dramatically altered (thereby enhancing adherence). It has to be mentioned that the receptor for the type 1 pilus and its adhesin RrgA is still unknown [43].

Pneumococci have to evade the host immune system for successful colonization and invasion. Macrophages are evolutionarily ancient members of the innate immune system and serve to eliminate microorganisms by phagocytosis. Capsule production directly prevents phagocytosis, and the capsule is therefore the *sine qua non* virulence factor. Several TCSs have already been shown to regulate proteins contributing to immune evasion: (i) TCS09 influences capsule production in D39 [23], (ii) TCS02 regulates PspA [44] and (iii) TCS06 controls PspC expression [45]. Thus, phagocytosis assays were performed to determine the impact of TCS09 on the immune evasion potential of TIGR4 pneumococci. Proteins contributing to immune evasion like CbpL, Ply and PspC [46,47,48,49] were not differentially produced in *tcs09*-mutants, as shown by immunoblot analysis (Figure 2). However, a multitude of proteins are involved in immune evasion and resistance against phagocytosis including, e.g., ClpP, ZmpC and EndA [50,51,52]. Immunofluorescence microscopy demonstrated that uptake of the pneumococcal *tcs09*-mutant strains by macrophages was similar to the wild-type (Figure 5A,B). In addition, the number of recovered wild-typeTIGR4Δ*cps* and *tcs09*-mutants was also unchanged (Figure 5C). In a following approach, we have further tested whether the loss of function of the TCS09 impairs intracellular survival of pneumococci in macrophages. Intracellular pneumococci are confronted with a harsh environment and especially reactive oxygen species (ROS) and acidic conditions. Our earlier studies with D39 indicated a higher sensitivity of *tcs09*-mutants against hydrogen peroxide [23]. In this study, a significant lower number of viable pneumococci deficient in the complete TCS09 was recovered 3 h after *S. pneumoniae* TIGR4 phagocytosis (Figure 5D). This effect was significant but not dramatic, suggesting only a minor influence of TCS09 on the intracellular fate of TIGR4. Other proteins that likely contribute to the survival in the macrophages after phagocytosis include (i) the arginine-deaminase-system (*arc*-operon), which produces ammonium and contributes to tolerate the low pH in phagolysosomes [14,53,54]; and (ii) other proteins like Etrx1, Etrx2 and SodA, whose expression is unchanged in *tcs09*-mutants, and NADH oxidase, PsaBCA and HtrA as well. The latter proteins are involved in defense against oxidative stress [29,55,56,57]. However, all published data and this study showed no differential gene or protein expression for these virulence candidates, which is in accordance with the intracellular behavior of our *tcs09*-mutants.

We further conducted in vivo infection experiments because all in vitro studies can only partially mirror the complex in vivo conditions, such as nutrient availability, oxygen level, temperature and attack by the host immune system. In the mouse infection models of acute pneumonia and septicemia, all *tcs09*-mutants were as virulent as the parental strain TIGR4*lux* (Figure 6B and Figure 7). In contrast to our results are virulence studies, in which the deficiency of RR09 in TIGR4 attenuated the mutant. The route of infection was identical, whereas different outbred mice were used [11,25]. Based on these conflicting data, we extended our in vivo experiments to investigate the impact of TCS09 on virulence in a competitive mouse infection model.

Here, the infection doses of the parental and mutant strain co-infected in a mouse are identical. This model is used to show the higher bacterial fitness by superiority over the other strain in a defined host compartment. Lower bacterial loads of *tcs09*-mutants were detected mainly in the bronchoalveolar lavage, lungs, blood and brain (Figure 8), while the CFU of *tcs09*-mutants and wild-type in nasopharynx was similar, as indicated by median CI values between 0.86 and 1.25. The mouse co-infection experiments clearly indicate that the TCS09 is crucial for full virulence of pneumococci, a finding that is probably independent of the genetic background of *S. pneumoniae*. However, the degree of attenuation is probably a strain-specific matter, which explains the different results in the TCS09 studies published so far. This in turn is most likely linked to the strain-specific regulation of fitness and virulence factors. Because the environmental signal sensed by HK09 is still unknown, one can only speculate under what conditions TCS09 is switched on and required for pneumococci during colonization or under infection-relevant conditions.

Moreover, we cannot exclude cross-talk between individual pneumococcal TCSs or compensation mechanisms, making it yet impossible to accurately define the role of TCS09 during pneumococcal pathogenesis. It will be interesting to analyze the in vivo transcriptome and proteome of RR09-, HK09- and TCS09-deficient pneumococci recovered from different host compartments to identify the genes targeted by the RR09 of TCS09.

## 5. Conclusions

In conclusion, these data confirm that TCS09 of TIGR4 and other pneumococcal strains are most likely not directly involved in regulation of factors essential for pneumococcal virulence. Instead, TCS09 seems to be crucial to maintain metabolic fitness and resistance under in vivo conditions, thereby facilitating dissemination and survival in different host compartments. The orchestrated activities of regulators enable pneumococci to survive and spread within the host. However, the specific impact of TCS09 on these processes and the environmental signal that triggers TCS09 needs to be elucidated in future studies.

## Figures and Tables

**Figure 1 microorganisms-09-01365-f001:**
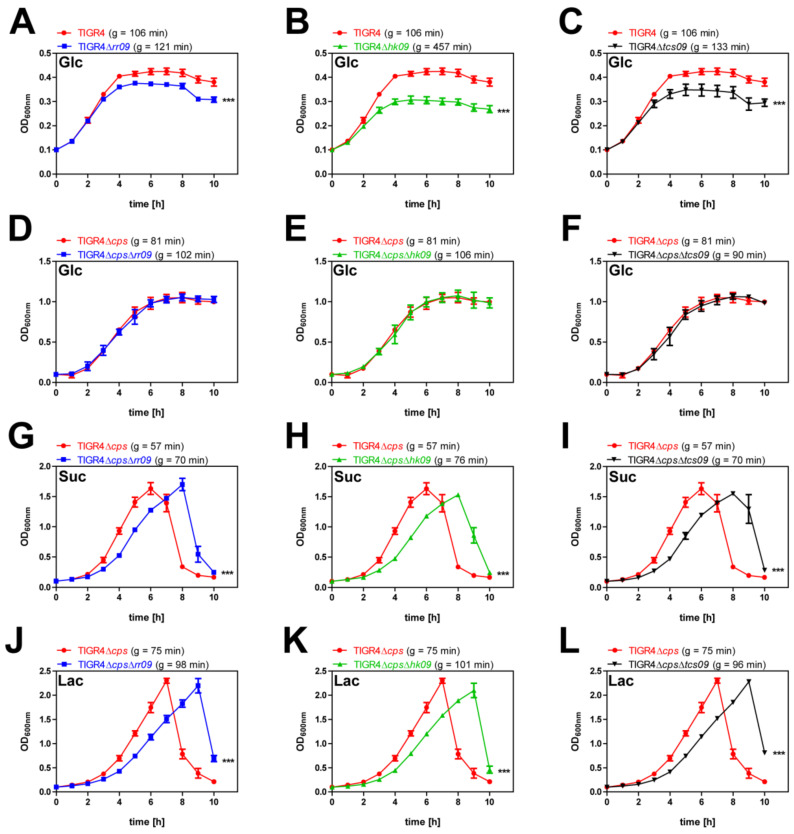
Growth analyses of *S. pneumoniae* TIGR4, TIGR4Δ*cps* and isogenic *tcs09*-mutants in CDM. Growth of encapsulated (**A**–**C**) and nonencapsulated (**D**–**L**) *tcs09*-mutants compared to the corresponding parental wild-type strains TIGR4 or TIGR4Δ*cps* was analyzed under nutrient-defined conditions with glucose (Glc) (**A**–**F**), sucrose (Suc) (**G**–**I**) and lactose (Lac) (**J**–**L**) as sole carbon source at 37 °C under microaerophilic conditions without agitation. Results are presented as the mean ± SD of three independent experiments. The mean value of the doubling time (g) from three biological replicates of the respective strain is given in brackets. A two-way Anova proved significance with a *p*-value *** < 0.001 relative to the parental pneumococcal strain.

**Figure 2 microorganisms-09-01365-f002:**
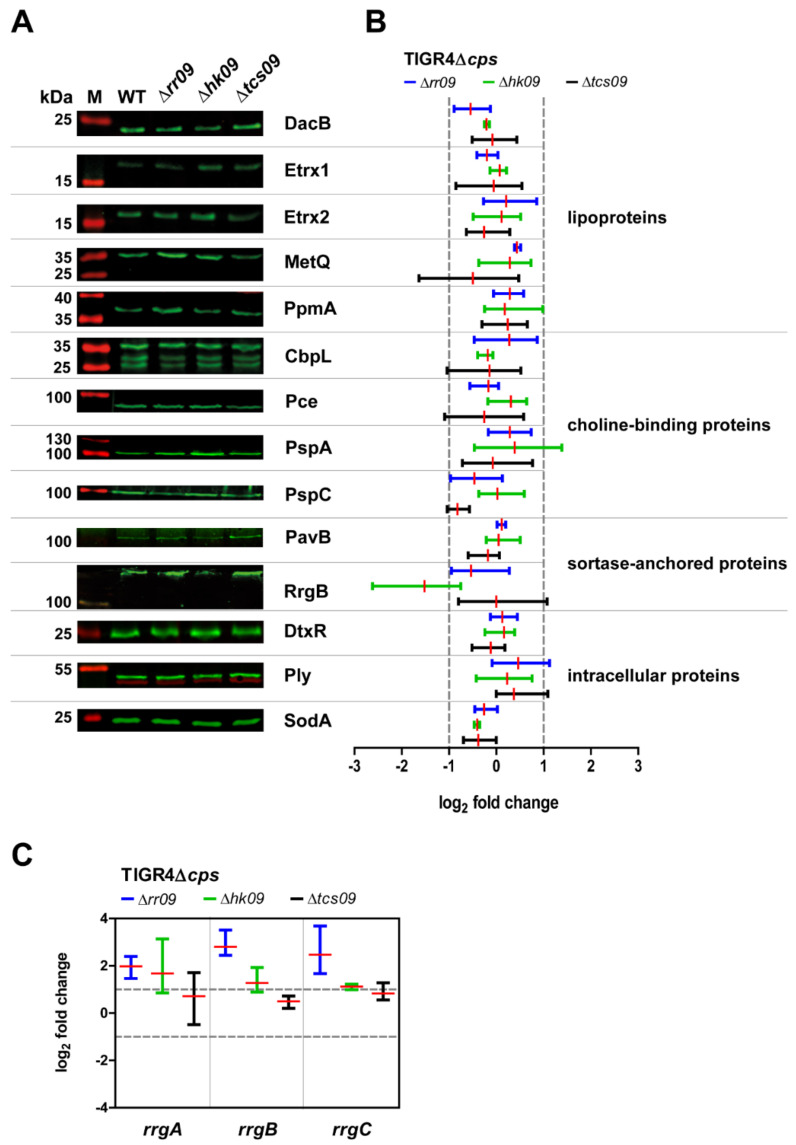
Immunoblot analysis of pneumococcal virulence factors. Pneumococcal strains were cultured to mid-exponential growth phase, and total cell lysates of 2 × 10^8^ bacteria were separated in a SDS-PAGE and plotted on nitrocellulose. Expression of indicated pneumococcal proteins was analyzed by immunoblot analysis using antigen-specific polyclonal mouse IgG and a secondary fluorescently labeled IRDye^®^ 800CW goat anti-mouse IgG antibody. A specific rabbit polyclonal anti-enolase IgG and a secondary fluorescent-labeled IRDye^®^ 680RD goat anti-rabbit IgG antibody served as normalization control. Shown are representative immunoblot images (**A**) and the differential expression pattern as log_2_ fold changes of representative candidates of lipoproteins, choline-binding proteins, sortase-anchored proteins and intracellular proteins (**B**). A log_2_ fold change < −1 and > 1 was set as significant for differential protein expression. (**C**) Differential gene expression of Pilus-1 components *rrgA*, *rrgB* and *rrgC* in *S. pneumoniae* TIGR4Δ*cps* and isogenic *tcs09*-mutants using qPCR. Pneumococci were grown in CDM with glucose to OD_600nm_ of 0.6; the RNA was isolated and reverse-transcribed into cDNA. A log_2_ fold change < −1 and > 1 was set as significant for differential gene expression. The log_2_ fold changes of differential gene expression in three replicates and their means are presented.

**Figure 3 microorganisms-09-01365-f003:**
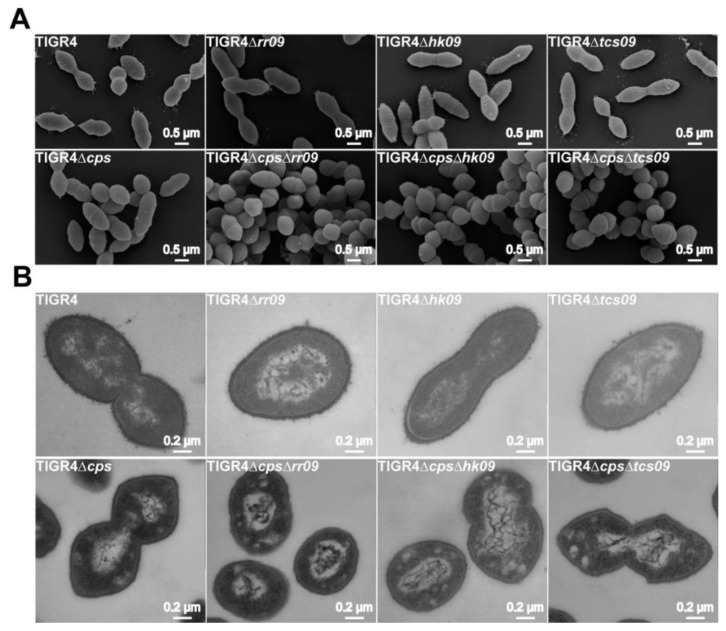
Illustration of the pneumococcal cell morphology in TCS09-deficient mutants using FESEM and TEM. Parental strains TIGR4, TIGR4Δ*cps* and isogenic *tcs09*-mutants were cultured in glucose-supplemented CDM. Pneumococci were subsequently fixed with paraformaldehyde and glutardialdehyde. The capsule of encapsulated pneumococci was preserved with a lysine-ruthenium-red solution. White bars in FESEM images (**A**) correspond to 500 nm and in TEM images (**B**) to 200 nm.

**Figure 4 microorganisms-09-01365-f004:**
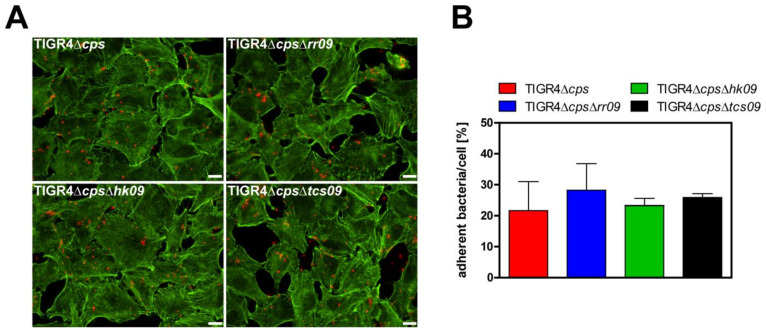
Adherence of *S. pneumoniae* TIGR4Δ*cps* and isogenic *tcs09*-mutants to A549 epithelial cells. Approximately 2 × 10^5^ A549 lung epithelial cells were infected with a MOI 25 of TIGR4Δ*cps* or isogenic *tcs09*-mutants for 3 h. Host-cell-bound pneumococci were labeled using polyclonal anti-pneumococcal antibodies and secondary antibody Alexa Fluor 568. After cell permeabilization, the actin cytoskeleton was stained using Phalloidin-iFluor™-488 conjugate. (**A**) Representative fluorescence images of adherent pneumococci on human lung epithelial cells A549 with a magnification of 630×. White bar represents 10 μm. (**B**) Adherent pneumococci of at least 50 cells/coverslip were counted via immunofluorescence microscopy. The results of three independent experiments are given as normalized percentage by relating the counted bacteria to the MOI set to 100%. Statistical analysis was performed with an unpaired *t*-test and revealed no significance.

**Figure 5 microorganisms-09-01365-f005:**
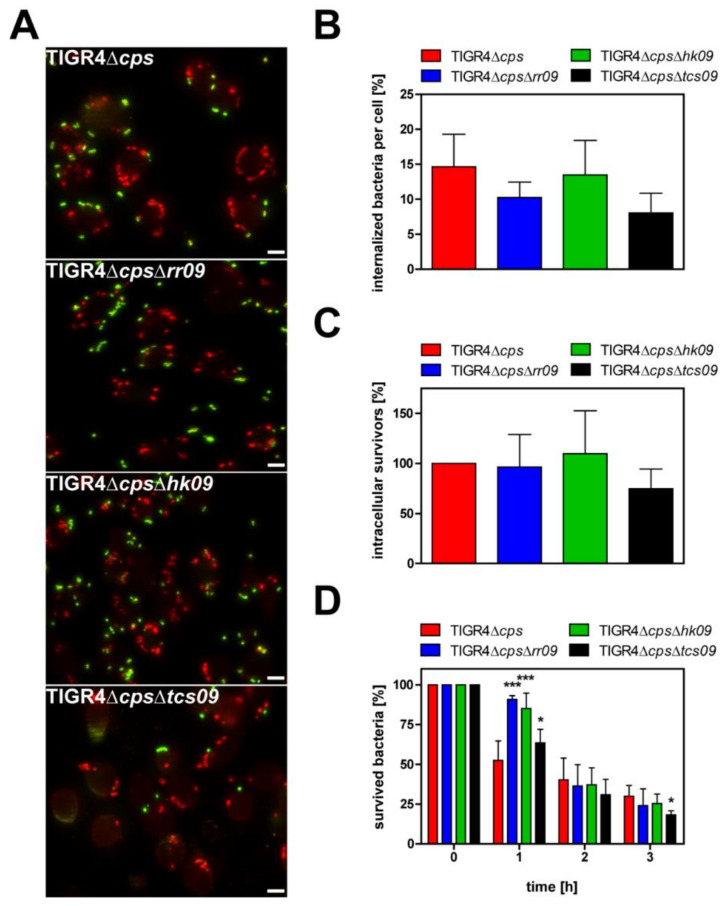
Uptake of *S. pneumoniae* TIGR4Δ*cps* and isogenic *tcs09*-mutants by murine macrophages. Approximately 2 × 10^5^ J774 cells were infected for 30 min with TIGR4Δ*cps* or isogenic *tcs09*-mutants with a MOI 50. Experiments were performed at least three times in triplicates. (**A**) Representative immunofluorescence images of pneumococcal uptake after 30 min of infection by double-immunofluorescence staining. Extracellular bacteria were labeled with primary polyclonal anti-pneumococcal antibody followed by secondary Alexa Fluor 488 coupled antibody (green) and intracellular bacteria with primary antibody and secondary Alexa Fluor 568 coupled antibody (red). Visualization was performed using a fluorescence microscope at 630× magnification. White bar represents 10 μm. (**B**) Intracellular pneumococci per cell in percentage of the inoculum. At least 50 cells were counted per coverslip to quantify the number of intracellular bacteria. (**C**) Intracellular pneumococcal survivors post incubation of macrophages and killing of extracellular bacteria by antibiotics. Intracellular pneumococci were recovered by permeabilization of J774 cells, and CFUs were determined by plating the cell lysates on blood agar. CFUs of wild-type TIGR4Δ*cps* were set to 100%. Data were normalized against the MOI. (**D**) Intracellular survival in macrophages over a time period of 3 h. After killing extracellular bacteria with antibiotics, infected macrophages were further incubated in infection medium in the absence of antibiotics. Numbers of intracellular pneumococci were determined by lysing J774 cells at different time points and plating the lysates on blood agar plates. Data were normalized against the number of bacteria isolated directly after antibiotic treatment (time point 0). A two-way Anova revealed a significance with *p*-value * < 0.05 and *** < 0.001 relative to the parental pneumococcal strain at the indicated time points.

**Figure 6 microorganisms-09-01365-f006:**
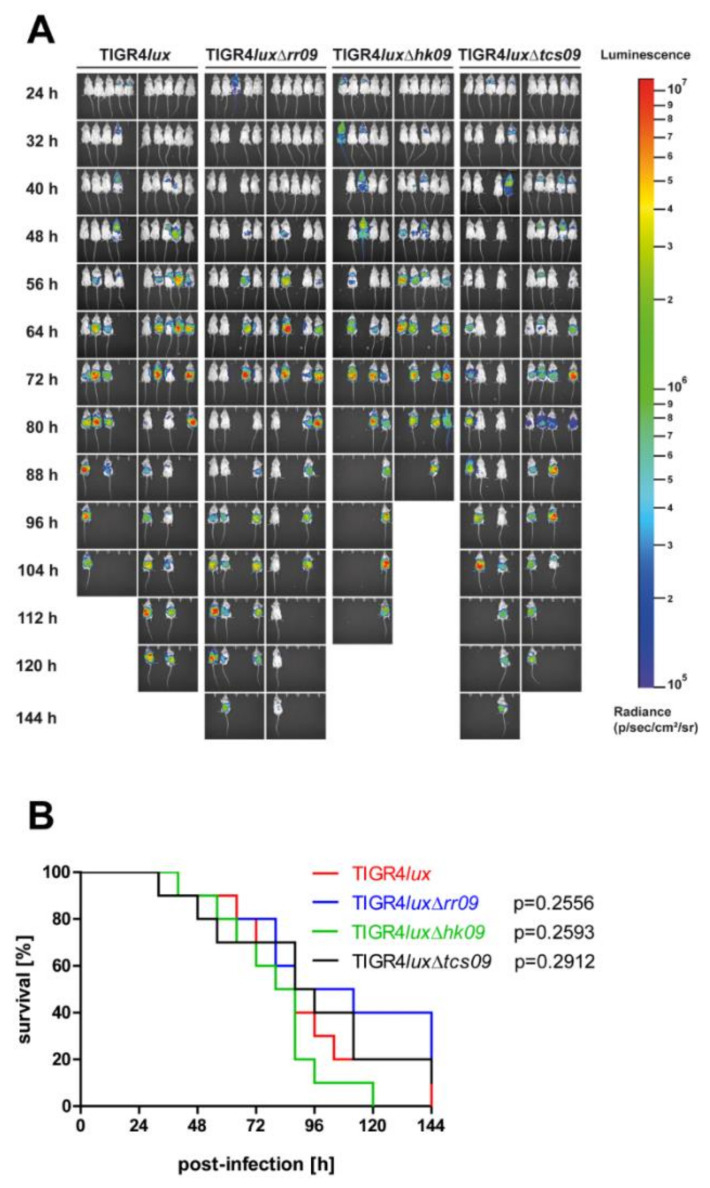
Survival of TIGR4 TCS09 deletion mutants in the acute pneumonia mouse infection model. CD-1 mice (*n* = 10) were intranasally infected with a CFU of 1 × 10^8^ bacteria of TIGR4*lux*, TIGR4*lux*Δ*rr09*, TIGR4*lux*Δ*hk09* or TIGR4*lux*Δ*tcs09* per mouse. (**A**) Spread of bioluminescent pneumococci was visualized at specific time points of infection by measurement of luminescence intensity (photons/second) using the IVIS^®^ Spectrum system. (**B**) Survival rates of infected mice were illustrated in a Kaplan–Meier diagram. Statistical analysis was performed with a log rank test and revealed no significance.

**Figure 7 microorganisms-09-01365-f007:**
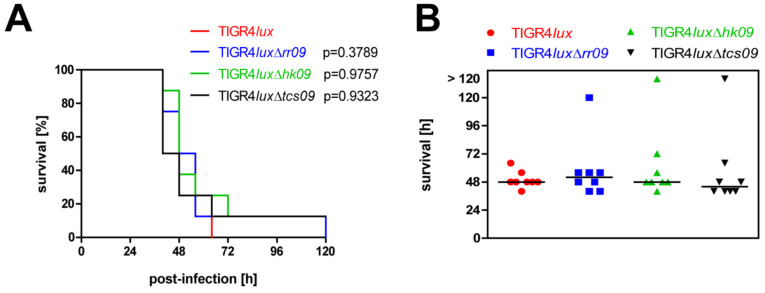
Survival of mice in a systemic mouse infection model. CD-1 mice (*n* = 8) were intraperitoneally infected with a CFU of 1 × 10^4^ of *S. pneumoniae* strains TIGR4*lux*, TIGR4*lux*Δ*rr09*, TIGR4*lux*Δ*hk09* and TIGR4*lux*Δ*tcs09*. (**A**) Kaplan–Meier diagram of surviving mice after intraperitoneal infection. Statistical analysis was performed with a log rank test and revealed no significance. (**B**) Scatter plot showing individual and median survival times per infected mouse group.

**Figure 8 microorganisms-09-01365-f008:**
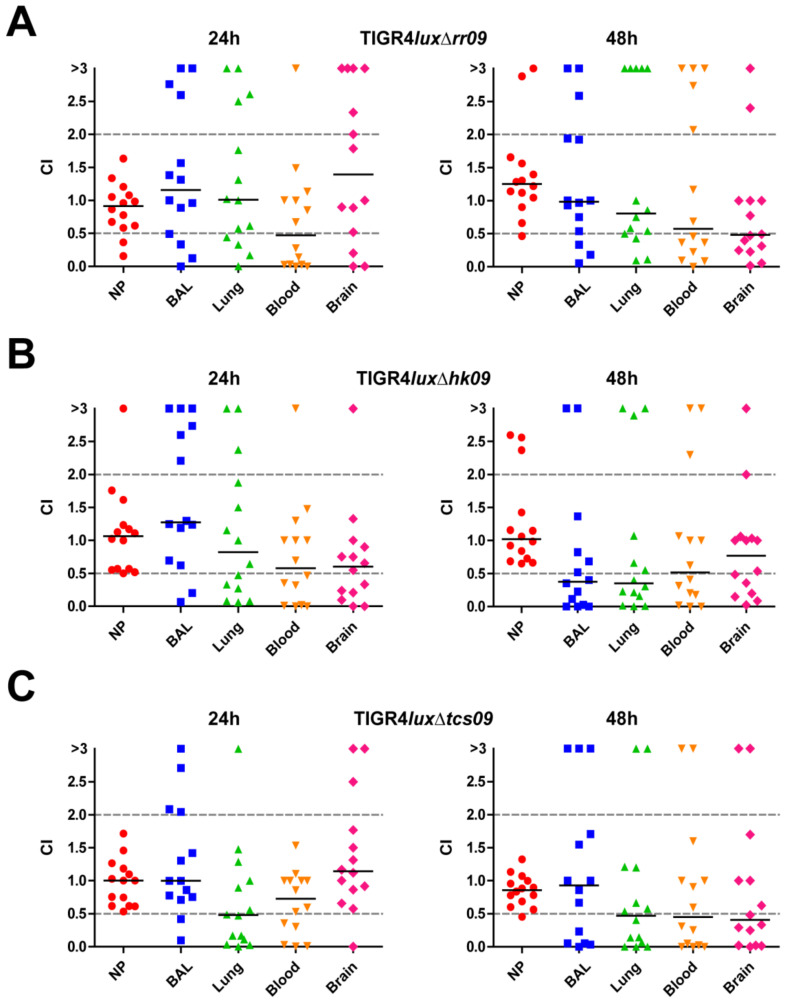
Intranasal co-infection of mice with bioluminescent TIGR4*lux* and isogenic *tcs09*-mutants. Three groups of CD-1 mice (*n* = 14) were infected with 1 × 10^7^ CFU of wild-type TIGR4*lux* together with 1 × 10^7^ CFU of one of the isogenic *tcs09*-mutants. At indicated time points (24 h or 48 h), mice were sacrificed and bacterial loads were counted in the nasopharynx (NP), bronchi (BAL: bronchoalveolar lavage), lung tissue, blood and brain after plating the recovered bacteria on blood agar plates. Shown are the CI values with median of the intranasal co-infection with (**A**) TIGR4*lux* and TIGR4*lux*Δ*rr09*, (**B**) TIGR4*lux* and TIGR4*lux*Δ*hk09* and (**C**) TIGR4*lux* and TIGR4*lux*Δ*tcs09*. CI values < 1 indicate a higher growth of wild-type bacteria than the respective mutant.

**Table 1 microorganisms-09-01365-t001:** *S. pneumoniae* wild-type strains and mutants used in this study.

Strain	Capsule Type	Resistance	Knockout Genes	Reference
TIGR4	4	-	-	[27]
TIGR4Δ*rr09*	4	erythromycin	*sp_0661*	This study
TIGR4Δ*hk09*	4	erythromycin	*sp_0662*	This study
TIGR4Δ*tcs09*	4	erythromycin	*sp_0661, sp_0662*	This study
TIGR4*lux*	4	kanamycin	-	[14]
TIGR4*lux*Δ*rr09*	4	kanamycin, erythromycin	*sp_0661*	This study
TIGR4*lux*Δ*hk09*	4	kanamycin, erythromycin	*sp_0662*	This study
TIGR4*lux*Δ*tcs09*	4	kanamycin, erythromycin	*sp_0661, sp_0662*	This study
TIGR4Δ*cps*	4	kanamycin	*sp_0343–sp_0360*	[14]
TIGR4Δ*cps*Δ*rr09*	4	kanamycin, erythromycin	*sp_0343–sp_0360, sp_0661*	This study
TIGR4Δ*cps*Δ*hk09*	4	kanamycin, erythromycin	*sp_0343–sp_0360, sp_0662*	This study
TIGR4Δ*cps*Δ*tcs09*	4	kanamycin, erythromycin	*sp_0343–sp_0360, sp_0661, sp_0662,*	This study

**Table 2 microorganisms-09-01365-t002:** Primers used for qPCR.

Target Gene	Primer	Sequence 5′–3′
*enolase* (*sp_1128*)	*eno*RT_F *eno*RT_R	CGGACGTGGTATGGTTCCA TAGCCAATGATAGCTTCAGCA
*rrgA* (*sp_0462*)	RT_*rrgA*_F RT_*rrgA*_R	GTGATTAAGGAGACAGGCGA GGTGTATGTCCCAGGTTTTAT
*rrgB* (*sp_0463*)	*rrgB*F2 *rrgB*mut2	TGGGACGACAACAACATC GATCAATATTCACTCCTAGAG
*rrgC* (*sp_0464*)	RT_*rrgC*_F RT_*rrgC*_R	CGGTTGCAAGTATGGAAGTT TACTTCAATCTGATTCTCAAGG

## Data Availability

Not applicable.

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
