# Peer review of "The Two-Component System 09 of Streptococcus pneumoniae Is Important for Metabolic Fitness and Resistance during Dissemination in the Host"

_microorganisms, 2021, doi:10.3390/microorganisms9071365_

Round 1
Reviewer 1 Report
In this study authors analyzed the role of Streptococcus pneumoniae two-component system (TCS09) in the strain TIGR4. Authors show some growth impairment with tcs09 mutants depending on the state of encapsulation and carbon source used in experiments. Furthermore, authors found that, tcs-09-mutants of TIGR4 do not show any gross changes in cell morphology and expression of several virulence related genes was unaffected. However, interestingly authors find that the expression of pilus backbone RgrB protein is reduced in hk09-mutant strain. Similarly in co-infection experiments, authors found that the TCS09 could be crucial for full virulence.
The experimental design and data presented are logical with proper interpretation. The quality of Figures is nice. However certain points need to be addressed as listed below.
Main suggestion:
- Do authors have any data from transcriptosome/Northern blotting analysis showing alterations in the expression of the rgrB mRNA with hk09- It will be nice to correlate changes. Such data should be presented.
- As mainly changes in the expression of RgrB protein are observed as presented in the Figure 2, it is imperative to present a nicer immunoblot. Bands corresponding to RgrB are diffuse. This panel needs to be replaced with sharper image.
- Is the decreased expression of Etrx1 in hk09 and tcs09- mutants (Figure 2) significant and why no change is observed in rr09-mutant? Authors should address it, although it is stated in page 17 Discussion section that no significant changes were observed. This needs clarification.
Author Response
Authors response to referee’s comments microorganisms-1253919
Manuscript title:
The two-component system 09 of Streptococcus pneumoniae is important for metabolic fitness and resistance during dissemination in the host
Reviewer comments and authors point-by-point response:
Reviewer 1
Comment 1:
Do authors have any data from transcriptosome/Northern blotting analysis showing alterations in the expression of the rgrBmRNA with hk09- It will be nice to correlate changes. Such data should be presented.
Response:
We thank the reviewer for this suggestion and have performed qPCR analysis to quantify expression of rrgA, rrgB and rrgC in all tcs09-mutants. The results are now included in Figure 2 as “C” and described in the results section as followed:
“To validate the differential expression of RrgB and Pilus-1 expression in TIGR4ΔcpsΔhk09, we applied qPCR. We therefore isolated RNA from TIGR4Δcps wild-type bacteria and isogenic tcs09-mutants grown in CDM with glucose as carbon source. The qPCR results revealed a significant upregulation of rrgA, rrgB and rrgC in rr09- and hk09-mutants, which is not in accordance to our protein expression analysis (Figure 2C). Thus, while the TCS09 seems to have a regulatory effect on Pilus-type 1 expression, the regulatory effect remains still unclear due to the difference in data at the protein level and transcriptome level.”
Figure 2. (C) Differential gene expression of Pilus-1 components rrgA, rrgB and rrgC in S. pneumoniae TIGR4Δcps and isogenic tcs09-mutants using qPCR. Pneumococci were grown in CDM with glucose to OD600nm of 0.6; the RNA was isolated and reverse transcribed into cDNA. A log2fold change < -1 and > 1 was set as significant for differential gene expression. The log2 fold changes of differential gene expression in three replicates and their means are presented.
These results were also discussed in the context of other studies (page 19; see labeled R1 version)
In addition, the Material and Method section was changed accordingly (page 3 and 4; see labeled R1 version). A table with primers is included as well (page 4)
Comment 2:
As mainly changes in the expression of RgrB protein are observed as presented in the Figure 2, it is imperative to present a nicer immunoblot. Bands corresponding to RgrB are diffuse. This panel needs to be replaced with sharper image.
Response:
The reviewer is addressing an important but also complicated point. The RrgB protein is difficult to detect in immunoblot in high quality. Unfortunately, we are not able to generate an immunoblot in a perfect and high resolution quality. The system which we are using for immunoblot analysis is the only one which allows a real quantitative analysis, while this is not perfectly applicable for ECL blots. We have repeated the immunoblots with anti-RrgB polyclonal antibodies and secondary fluorescent labeled IRDye® 800CW (goat-anti-mouse IgG; 1:15,000 in 5% skim milk in TBS/0.01% Tween) many times. Figure 2A contains a representative blot of our RrgB blots and is one of the best immunoblots we got. We tried to get a better blot (see files for reviews), but were not successful to improve the immunoblot. Also other studies showing RrgB blots had similar problems. Therefore we would like to leave Figure 2A as it is.
Comment 3:
Is the decreased expression of Etrx1 in hk09 and tcs09- mutants (Figure 2) significant and why no change is observed in rr09-mutant? Authors should address it, although it is stated in page 17 Discussion section that no significant changes were observed. This needs clarification.
Response:
We apologize for unclear information regarding Etrx1 expression in the tcs09-mutants. We defined a log2 fold change < -1 and > 1 as significant for differential protein expression. With log2 fold changes between -0.86 and 0.54 we could not detect any significant differential Etrx1 expression between tcs09-mutants and the wild-type TIGR4Δcps. That’s why we stated “unchanged expression in tcs09-mutants” in the Discussion section.
In addition, Figure legend of 2B was completed: “A log2 fold change < -1 and > 1 was set as significant for differential protein expression.”
Reviewer 2
Comments: none
Response:
The authors appreciate the positive evaluation of our study
Reviewer 2 Report
In my opinion, the work of Hirschmann S. and co-workers is well designed and the experiments were carried out carefully. The introduction is well structured, with the correct inputs for the readers. The materials and methods section, explains very well the experimental procedures for all the tests conducted. The results are well presented and discussed. Thus, in my opinion, the paper is already available for publication.
Author Response
Reviewer 2
Comments: none
Response:
The authors appreciate the positive evaluation of our study